# Modeling Uncertainty by Learning a Hierarchy of Deep Neural Connections

**Raanan Y. Rohekar**
Intel AI Lab
raanan.yehezkel@intel.com

**Yaniv Gurwicz**
Intel AI Lab
yaniv.gurwicz@intel.com

**Shami Nisimov**
Intel AI Lab
shami.nisimov@intel.com

**Gal Novik**
Intel AI Lab
gal.novik@intel.com

## Abstract

Modeling uncertainty in deep neural networks, despite recent important advances, is still an open problem. Bayesian neural networks are a powerful solution, where the prior over network weights is a design choice, often a normal distribution or other distribution encouraging sparsity. However, this prior is agnostic to the generative process of the input data, which might lead to unwarranted generalization for out-of-distribution tested data. We suggest the presence of a confounder for the relation between the input data and the discriminative function given the target label. We propose an approach for modeling this confounder by sharing neural connectivity patterns between the generative and discriminative networks. This approach leads to a new deep architecture, where networks are sampled from the posterior of local causal structures, and coupled into a compact hierarchy. We demonstrate that sampling networks from this hierarchy, proportionally to their posterior, is efficient and enables estimating various types of uncertainties. Empirical evaluations of our method demonstrate significant improvement compared to state-of-the-art calibration and out-of-distribution detection methods.

## 1 Introduction

Deep neural networks have become an important tool in applied machine learning, achieving state-of-the-art regression and classification accuracy in many domains. However, quantifying and measuring uncertainty in these discriminative models, despite recent important advances, is still an open problem. Representation of uncertainty is crucial for many domains, such as safety-critical applications, personalized medicine, and recommendation systems [6]. Common deep neural networks are not designed to capture model uncertainty, hence estimating it implicitly from the prediction is often inaccurate. Several types of uncertainties are commonly discussed [5, 19], where the two main types are: 1) epistemic uncertainty and 2) aleatoric uncertainty. Epistemic uncertainty is caused by the lack of knowledge, typically in cases where only a small training data exists, or for out-of-distribution inputs. Aleatoric uncertainty is caused by noisy data, and contrary to epistemic uncertainty, does not vanish in the large sample limit. In this work, we focus on two aspects of uncertainty, often required addressing in practical uses of neural networks: *calibration* [2] and *out-of-distribution* (OOD) detection.

Calibration is a notion that describes the relation between a predicted probability of an event and the actual proportion of occurrences of that event. It is generally measured using (strictly) *proper*

*scoring rules* [7], such as negative log-likelihood (NLL) and the Brier score, both are minimized for calibrated models. Guo et al. [8] examined the calibration of recent deep architectures by computing the expected calibration error (ECE)—the difference between an approximation of the empirical reliability curve and the optimal reliability curve [21]. Miscalibration is often addressed by post processing the outputs [24, 8], approximating the posterior over the weights of a pre-trained network [25], or by learning an ensemble of networks [13, 18, 5, 16].

OOD detection is often addressed by designing a loss function to optimize during parameter learning of a given network architecture [19, 3]. Many of these methods are specifically tailored for detecting OOD, requiring some information about OOD samples, which is often impractical to obtain in real-world cases. In addition, these methods often are not capable of modeling different types of uncertainty. Ensemble methods, on the other hand, learn multiple sets of network parameters for a given structure [16], or approximate the posterior distribution over the weights from which they sample at test time [1, 18, 5].

In this paper, we make the distinction between *structure-based* methods, which include ensemble methods that replicate the structure or sample subsets from it, and *parameter-based* methods, which specify a loss function to be used for a given structure. Ensemble methods, in general, do not specify the loss function to be used for parameter learning, nor do they restrict post-processing of their output. It is interesting to note that while the majority of ensemble methods use distinct sets of parameters for each network [16], in the MC-dropout method [5] the parameters are shared across multiple networks. Common to all these methods is that they use a single network architecture (structure), as it is generally unclear how to fuse outputs from different structures.

We propose a method that samples network structures, where parts of one sampled structure may be similar to parts of another sampled structure but having different weights values. In addition, these structures may share some parts, along with their weights, with other structures (weight sharing), specifically in the deeper layers. All these properties are learned from the input data.

## 2 Background

We focus on two approaches that are commonly used for modeling uncertainty: 1) Bayesian neural networks [5, 1], and 2) ensembles [16, 18]. Both approaches employ multiple networks to model uncertainty, where the main difference is the use/lack-of-use of shared parameters across networks during training and inference.

In Bayesian neural networks the weights, $\phi$, are treated as random variables, and the posterior distribution is learned from the training data $p(\phi|\mathbf{x}, \mathbf{y})$. Then, the probability of label $y^*$ for a test sample $x^*$ is evaluated by

$$p(y^*|\boldsymbol{x}^*, \mathbf{x}, \mathbf{y}) = \int p(y^*|\boldsymbol{x}^*, \phi)\, p(\phi|\mathbf{x}, \mathbf{y})\, d\phi. \tag{1}$$

However, since learning the posterior over $\phi$ and estimating Equation 1 are usually intractable, variational methods are often used, where an approximating variational distribution is defined $q(\phi)$, and the KL-divergence between the true posterior and the variational distribution, $\mathrm{KL}(q(\phi)\,||\,p(\phi|\boldsymbol{X}, Y))$, is minimized.

A common practice is to set a prior $p(\phi)$ and use the Bayes rule,

$$p(\phi|\boldsymbol{X}, Y) = \frac{p(Y|\boldsymbol{X}, \phi)\, p(\phi)}{p(Y|\boldsymbol{X})}. \tag{2}$$

Thus, minimizing the KL-divergence is equivalent to maximizing a variational lower bound,

$$\mathcal{L}_{\mathrm{VI}} = \int q(\phi) \log\left[p(Y|\boldsymbol{X}, \phi)\right] d\phi - \mathrm{KL}\Big(q(\phi)\,||\,p(\phi)\Big) d\phi. \tag{3}$$

Gal & Ghahramani [5] showed that the dropout objective, when applied before every layer, maximizes Equation 3. However, the prior $p(\phi)$ is agnostic to the unlabeled data distribution $p(\mathbf{x})$, which may lead to unwarranted generalization for out-of-distribution test samples. As a remedy, in this work we propose to condition the parameters of the discriminative model, $\phi$, on the unlabeled training data, $\boldsymbol{X}$. That is, to replace the prior $p(\phi)$ in Equation 2 with $p(\phi|\mathbf{x})$, thereby letting unlabeled training data to guide the posterior distribution rather than relying on some prior assumption over the prior.

## 3 A Hierarchy of Deep Neural Networks

We first describe the key idea, then introduce a new neural architecture, and finally, describe a stochastic inference algorithm for estimating different types of uncertainties.

### 3.1 Key Idea

We approximate the prediction in Equation 1 by sampling from the posterior, $\phi_i \sim p(\phi|\mathbf{x}, \mathbf{y})$,

$$P(y^*|\boldsymbol{x}^*, \mathbf{x}, \mathbf{y}) \approx \frac{1}{m} \sum_{i=1}^{m} P(y^*|\boldsymbol{x}^*, \phi_i). \tag{4}$$

Since sampling from the posterior is intractable for deep neural networks, we follow a Bayesian approach and propose a prior distribution for the parameters. However, in contrast to the common practice of assuming a Gaussian distribution (or some other prior independent of the data), our prior depends on the unlabeled training data, $p(\phi|\mathbf{x})$. We define $p(\phi|\mathbf{x})$ by first, considering a generative model for $\mathbf{x}$, with parameters $\theta$. Next, we assume a dependency relation between $\theta$ and $\phi$, such that the joint distribution factorizes as

$$p(X, Y, \phi, \theta) = p(Y|X, \phi)\, p(X|\theta)\, p(\phi|\theta)\, p(\theta). \tag{5}$$

In essence, we assume a generative process of $\boldsymbol{X}$, to confound the relation $\boldsymbol{X} \rightarrow \phi$ conditioned on $Y$. That is, in contrast to the common practice where a "v-structure" is assumed, $\boldsymbol{X} \rightarrow Y \leftarrow \phi$, we assume a generative function, parametrized by $\theta$, to be a parent of $\boldsymbol{X}$ and $\phi$, as illustrated in Figure 1. Given a training set $\{\mathbf{x}, \mathbf{y}\}$ the posterior is

$$p(\phi|\mathbf{x}, \mathbf{y}) \propto \int p(\mathbf{y}|\mathbf{x}, \phi)\, p(\phi|\theta)\, p(\theta|\mathbf{x})\, d\theta. \tag{6}$$

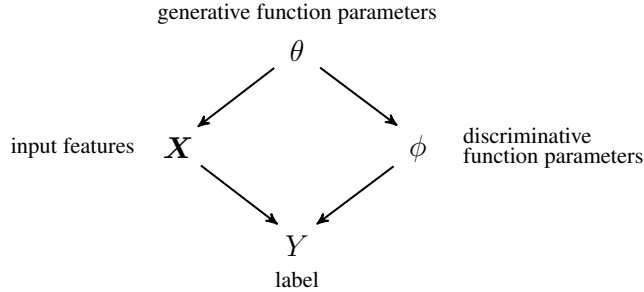

Figure 1: A causal diagram describing our assumptions.

The prior $p(\phi|\theta)$ is expected to diminish unwarranted generalization for out-of-distribution samples—samples with low $p(\boldsymbol{X}|\theta)$. As these samples are non-present or scarce in the training set, they have non or negligible influence on $\theta$. Hence, for in-distribution, $X_{\text{in}} \sim p(\mathbf{x})$, $p(\phi|\boldsymbol{X}_{\text{in}}, \theta) = p(\phi|\theta)$, whereas for out-of-distribution this relation does not hold. During inference we first sample $\theta$ from $p(\theta|\mathbf{x})$, which for an arbitrary out-of-distribution sample is a uniform distribution. Thus, $p(\phi|\theta)$ is expected to spread probability mass across $\phi$ for out-of-distribution data.

Two main questions arise: 1) what generative model should be used, and 2) how can we define the conditional relation $p(\phi|\theta)$. It is desired to define a generative model such that the conditional distribution $p(\phi|\theta)$ can be computed efficiently.

Our solution includes a new deep model, called BRAINet, which consists of multiple networks coupled into a single hierarchical structure. This structure (inter-layer neural connectivity pattern) is learned and scored from unlabeled training data, $\mathbf{x}$, such that multiple generative structures, $\{\tilde{\mathcal{G}}_1, \tilde{\mathcal{G}}_2, \tilde{\mathcal{G}}_3, \ldots\}$, can be sampled from their posterior $p(\tilde{\mathcal{G}}|\mathbf{x})$ during training and inference. We define $p(\theta_i|\mathbf{x}) \equiv p(\tilde{\mathcal{G}}_i|\mathbf{x})$, where for each $\theta_i$, the corresponding discriminative network parameters are estimated $\phi_i = \arg\max p(\phi|\theta_i)$. This can be described as using multiple networks during inference and training, similarly to MC-dropout [5] and Deep Ensembles [16]. However, as these structures are sampled from a single network, they share some of their parameters, specifically in deeper layers. This is different from MC-dropout where all the parameters are shared across networks, and Deep Ensembles where none of the parameters are shared.

## 3.2 BRAINet: A Hierarchy of Deep Networks

Recently, Rohekar et al. [27] introduced an algorithm, called B2N, for learning the structure, $\mathcal{G}$, of discriminative deep neural networks in an unsupervised manner. The B2N algorithm, learns an inter-layer connectivity pattern, where neurons in a layer may connect to other neurons in any deeper layer, not just to the ones in the next layer. Initially, B2N learns a deep generative graph $\tilde{\mathcal{G}}$ with latent nodes $\boldsymbol{H}$. This graph is constructed by unfolding the recursive calls in the RAI structure learning algorithm [30]. RAI learns a causal structure[1] $\mathcal{B}$ [23], and the B2N learns a deep generative graph such that,

$$p_{\mathcal{B}}(\boldsymbol{X}) = \int p_{\tilde{\mathcal{G}}}(\boldsymbol{X}, \boldsymbol{H}) d\boldsymbol{H}. \tag{7}$$

Interestingly, a discriminative network structure $\mathcal{G}$ is proved to mimic [23] a generative structure $\tilde{\mathcal{G}}$, having the exact same structure for $(\boldsymbol{X}, \boldsymbol{H})$. That is, for any $\theta'$, there exists $\phi'$, which can produce the posterior distribution over the latent variables in the generative model, $p_{\theta'}(\boldsymbol{H}|\boldsymbol{X}) = p_{\phi'}(\boldsymbol{H}|\boldsymbol{X}, Y)$[2].

Recently, an extension for the RAI algorithm [30] was proposed, called B-RAI [26]. B-RAI is a Bayesian approach that learns multiple causal structures, scores them and couples them into a hierarchy. This hierarchy is represented by a tree, which they call GGT. The Bayesian scores of each structure is encoded efficiently in the GGT, which allows structures $\{\mathcal{B}_1, \mathcal{B}_2, \ldots\}$ to be sampled from this GGT proportionally to their posterior distribution, $P(\mathcal{B}|\boldsymbol{X})$. Based on the principles of the B2N algorithm, we propose converting this GGT (generated by B-RAI) into a deep neural network hierarchy. We call this hierarchy B-RAI neural network, abbreviated to BRAINet. Then, a neural network structure, $\mathcal{G}$, can be sampled from the BRAINet model proportionally to $P(\mathcal{B}|\boldsymbol{X})$, where $\mathcal{G}$ has the same connectivity as a generative structure $\tilde{\mathcal{G}}$, and where the relation in Equation 7 holds. This yields a dependency between the generative $P(\boldsymbol{X})$ and discriminative $P(Y|\boldsymbol{X})$ models.

### 3.2.1 BRAINet Structure Learning

Before describing the BRAINet structure learning algorithm, we provide definitions of relevant concepts introduced by Yehezkel & Lerner [30].

**Definition 1** (Autonomous set of nodes). *In a graph defined over $\boldsymbol{X}$, a set of nodes $\boldsymbol{X}' \subseteq \boldsymbol{X}$ is called autonomous given $\boldsymbol{X}_{\mathrm{ex}} \subset \boldsymbol{X}$ if the parents' set, $\boldsymbol{Pa}(X)$, $\forall X \in \boldsymbol{X}'$ is $\boldsymbol{Pa}(X) \subset \boldsymbol{X}' \cup \boldsymbol{X}_{\mathrm{ex}}$.*

**Definition 2** (d-separation resolution). *The resolution of a d-separation relation between a pair of non-adjacent nodes in a graph is the size of the smallest condition set that d-separates the two nodes.*

**Definition 3** (d-separation resolution of a graph). *The d-separation resolution of a graph is the highest d-separation resolution in the graph.*

We present a recursive algorithm, Algorithm 1, for learning the structure of a BRAINet model. Each recursive call receives a causal structure $\mathcal{B}$ (a CPDAG), a set of endogenous $\boldsymbol{X}$ and exogenous $\boldsymbol{X}_{\mathrm{ex}}$ nodes, and a target conditional independence order $n$. The CPDAG encodes $P(\boldsymbol{X}|\boldsymbol{X}_{\mathrm{ex}})$, providing an efficient factorization of this distribution. The d-separation resolution of $\mathcal{B}$ is assumed $n-1$.

At the beginning of each recursive call, an exit condition is tested (line 2). This condition is satisfied if conditional independence of order $n$ cannot be tested (a conditional independence order is defined to be the size of the condition set). In this case, the maximal depth is reached and an empty graph is returned (a gather layer composed of observed nodes). From this point, the recursive procedure will trace back, adding latent parent layers.

Each recursive call consists of three stages:

    a) Increase the d-separation resolution of $\mathcal{B}$ to $n$ and decompose the input features, $\boldsymbol{X}$, into autonomous sets of nodes (lines 7–9): one descendant set, $\boldsymbol{X}_{\mathrm{D}}$, and $k$ ancestor sets, $\{\boldsymbol{X}_{\mathrm{A}i}\}_{i=1}^k$.

    b) Call recursively to learn BRAINet structures for each autonomous set (lines 10–12).

    c) Merge the returned BRAINet structures into a single structure (lines 13–15).

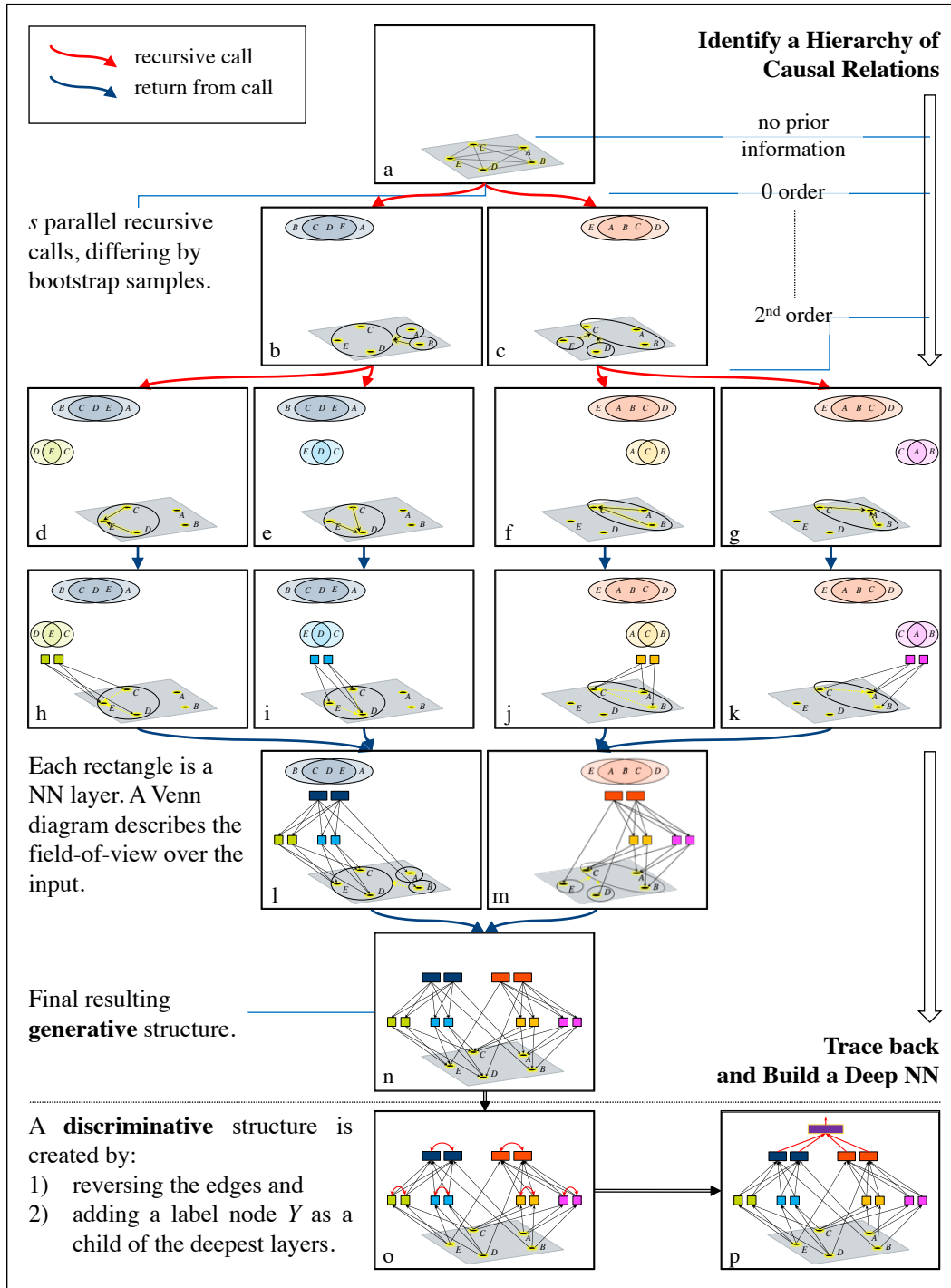

Figure 2: An example of BRAINet structure learning with $s = 2$. Network inputs, $\boldsymbol{X} = \{A, \ldots, E\}$, are depicted on a gray plain. Red arrows indicate recursive calls for learning local causal structures. Initially, no prior information exists and a fully-connected graph over $\boldsymbol{X}$ is assumed (a). Then, 0-order (condition set size of 0) independence is tested between pairs of connected nodes, using two ($s$) different bootstrap samples of the training data. This results in two CPDAGs (b, c). These CPDAGs are further refined by recursive calls with higher order independence tests, until no more edges can be removed (d-g). Tracing back, a deep NN is constructed by adding NN layers, where each CPDAG leads to its corresponding neural connectivity pattern (h-n). Each rectangle represents a layer of neurons (denoted $L_i^t$ in Algorithm 1-line 14). A Venn diagram, above a set of rectangles, indicates the field-of-view each layer has over $\boldsymbol{X}$. Finally, a discriminative structure is created (o, p).

---

**Algorithm 1:** BRAINet structure learning

1 **BRAINet_SL** $(\mathcal{B}, \boldsymbol{X}, \boldsymbol{X}_{\mathrm{ex}}, n)$
    **Input:** an initial CPDAG $\mathcal{B}$ over endogenous $\boldsymbol{X}$ & exogenous $\boldsymbol{X}_{\mathrm{ex}}$ observed variables, and a desired
          resolution $n$.
    **Output:** $L$, the deepest layer in a learned structure

2     **if** *the maximal indegree of $\boldsymbol{X}$ in $\mathcal{B}$ is lower than $n+1$* **then**          ▷ `exit condition`
3         $r := \mathtt{Score}(\boldsymbol{X}|\mathcal{B})$          ▷ `a Bayesian score (e.g., BDeu)`
4         $L :=$ a gather layer for $\boldsymbol{X}$ with score $r$
5         **return** $L$

6     **for** $t = 1 : s$ **do**
7         $\mathbf{x}^* :=$ sample with replacement from training data $\mathbf{x}$         ▷ `bootstrap sample`
8         $\mathcal{B}^* := \mathtt{IncSeparation}(\mathcal{B}, n, \mathbf{x}^*)$     ▷ `increase d-separation resolution to n`
9         $\{\boldsymbol{X}_{\mathrm{D}}, \boldsymbol{X}_{\mathrm{A}1}, \ldots, \boldsymbol{X}_{\mathrm{A}k}\} := \mathtt{FindAutonomous}(\boldsymbol{X}|\mathcal{B}^*)$     ▷ `decompose`
10         **for** $i = 1 : k$ **do**
11             $L_{\mathrm{A}i} := \mathtt{BRAI\_NN\_SL}(\mathcal{B}^*, \boldsymbol{X}_{\mathrm{A}i}, \boldsymbol{X}_{\mathrm{ex}}, n+1)$     ▷ `recursively call for ancestors`
12         $L_{\mathrm{D}} := \mathtt{BRAI\_NN\_SL}(\mathcal{B}^*, \boldsymbol{X}_{\mathrm{D}}, \boldsymbol{X}_{\mathrm{ex}} \cup \{\boldsymbol{X}_{\mathrm{A}i}\}_{i=1}^k, n+1)$ ▷ `recursively call for descendant`
13         Create an empty layer container $\boldsymbol{L}^t$ (tagged with index $t$)
14         In $\boldsymbol{L}^t$ create $k$ independent layers: $L_1^t, \ldots, L_k^t$
15         $\forall i \in \{1, \ldots, k\}$, connect: $L_{\mathrm{A}i} \rightarrow L_i^t \leftarrow L_{\mathrm{D}}$     ▷ `connect`
16     **return** $L = \{\boldsymbol{L}^t\}_{t=1}^s$

---

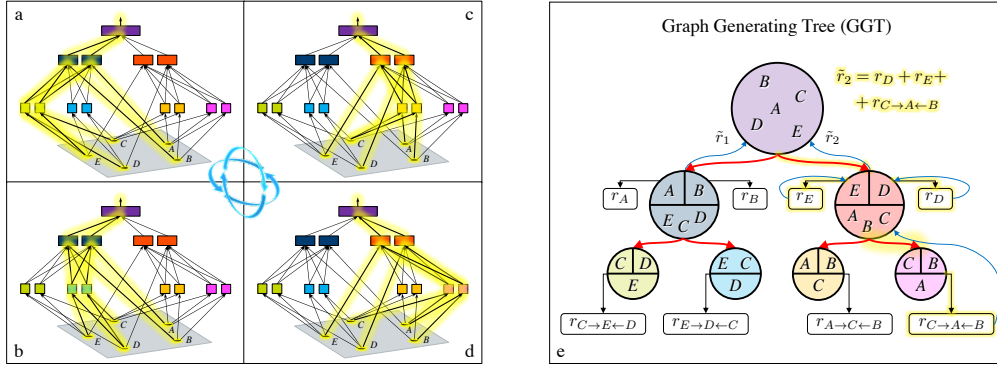

Figure 3: Stochastic training/inference in a BRAINet model. In a single stochastic/training step, a subset of the network is selected. In this example there are four possible sub-network selections: a,b,c, and d, encoded in a GGT (e). In the GGT, one of $s$ (here $s = 2$) branches, having scores $r_1, \ldots, r_s$, is sampled according to Equation 8. As an example, the highlighted branches were sampled resulting in sub-network d. Note that, in every possible stochastic step (a,b,c,d), all the inputs, $\boldsymbol{X} = A, \ldots, E$, are selected, and each input is selected only once.

These stages are executed $s$ times (line 6), resulting in an ensemble of $s$ BRAINet structures. The deepest layers, $\{\boldsymbol{L}^1, \ldots, \boldsymbol{L}^s\}$, of these structures are grouped together (line 16), while maintaining their index $1, \ldots, s$ in $L$, and returned. Note that the caller function treats this group as a single layer. A detailed description of Algorithm 1 can be found in Appendix A, and complexity analysis in Appendix B. An example for learning a BRAINet structure with $s = 2$ (each recursive call returns an ensemble of two BRAINet models) is given in Figure 2.

### 3.2.2 BRAINet Training and Inference

The BRAINet model allows us to sample sub-networks with respect to their relative posterior probability. The scores calculation and sub-network selection are performed recursively, from the leaves to the root. For each autonomous set, given $s$ sampled sub-networks and their scores, $r_1, \ldots, r_s$, returned from $s$ recursive calls, one of the $s$ results is sampled. We use the Boltzmann distribution,

$$P(t; \{r_{t'}\}_{t'=1}^s) = \frac{\exp[r_t/\gamma]}{\sum_{t'=1}^s \exp[r_{t'}/\gamma]}, \tag{8}$$

where $\gamma$ is a "temperature" term. When $\gamma \to \infty$, results are sampled from a uniform distribution, and when $\gamma \to 0$ the index of the maximal value is selected ($\arg\max$). We use $\gamma = 1$ and the Bayesian score, BDeu [10]. Finally, the sampled sub-networks, each corresponding to an autonomous set, are merged. The score of the resulting network is the sum of scores[3] of all autonomous sets merged into the network. When training the parameters, at each step, a single sub-network is sampled using a uniform distribution and its weights are updated. At inference, however, there are two options. In the the first, which we call "stochastic", we run $T$ forward passes, each time sampling a single network with respect to Equation 8. Then, the outputs of the sampled networks are averaged. Figure 3 illustrates several forward passes sampled from the BRAINet model. This is similar to dropout at inference, except that a sub-network is sampled with respect to its posterior. Note that, in BRAINet, in contrast to MC-dropout, weights are not sampled independently, and there is an implicit dependency between sampling variables. In the second inference option, which we call "simultaneous", we run a single forward pass through the BRAINet and recursively perform a weighted average of the $s$ activations for each autonomous set.

### 3.2.3 BRAINet Uncertainty Estimation

Several measures of uncertainty can be computed using the BRAINet model. Firstly, the max-softmax [28] and entropy [5] can be computed on the outputs of a single "simultaneous" forward pass or on the average of outputs from multiple stochastic forward passes. Secondly, using the distinct outputs of multiple forward passes, we can compute the expected entropy, $\mathbb{E}_{p(\phi|\mathbf{x},\mathbf{y})}\mathcal{H}[p(y^*|x^*,\phi)]$, or the mutual information, $\mathrm{MI}(y^*, \phi|x^*, \mathbf{x}, \mathbf{y})$, [29]. In Appendix C-Figure 2, we qualitatively show epistemic uncertainty estimation using MI, calculated from the BRAINet model, for images generated by VAE [15] trained on MNIST. In addition, using the outputs of multiple stochastic passes, we can estimate the distribution over the network output. Finally, we demonstrate an interesting property of our method that learns a broader prior over $\phi$ as the training set size decreases. That is, a relation between the predictive uncertainty and the number of unique structures (connectivity patterns) encoded in a BRAINet model (exemplified in Appendix C-Figure 1).

## 4  Empirical Evaluation

BRAINet structure learning algorithm is implemented using BNT [20] and runs efficiently on a standard desktop CPU. In all experiments, we used MLP-layers (dense), ReLU activations, ADAM optimization [14], a fixed learning rate, and batch normalization [12]. Unless otherwise stated, each experiment was repeated 10 times. Mean and STDev on test-set are reported. BRAINet structure was learned directly for pixels in the ablation study (Section 4.1) and calibration experiments (Section 4.2). In out-of-distribbution detection experiments, higher level features were extracted using the first convolutional layers of common pre-trained NN, and BRAINet structure was learned for these features.

BRAINet parameters were learned by sampling sub-networks uniformly, and updating the parameters by SGD. At inference, BRAINet performs Bayesian model averaging (each sampled sub-network is weighted) using one of the two strategies described in Section 3.2.2.

### 4.1  An Ablation Study for Evaluating the Effect of Confounding with a Generative Process

First, we conduct an ablation study by gradually reducing the dependence of the discriminative parameters $\phi$ on the generative parameters $\theta$, i.e., the strength of the link $\theta \to \phi$ in Figure 1, and measuring the performance of the resulting model for MNIST dataset [17]. In the extreme case of disconnecting this link, the BRAINet structure will simply become a Deep Ensembles model [16] with $s$ independent networks. Figure 4 demonstrates that even for a small dependence between $\theta$ and $\phi$, as restricted by BRAINet, a significant improvement is achieved in performance (high calibration and classification accuracy). $X$-axis represents the strength of the link $\theta \to \phi$ (see Figure 1), where the values represent the amount of mutual information that is required for a pair of nodes in $\boldsymbol{X}$ to be considered dependent (line 8, Algorithm 1). When this value is low, all the nodes in $\boldsymbol{X}$ are considered dependent and no structure is learned. For a mutual information threshold of 0, a simple ensemble of $s$ networks is obtained where each network is composed of stacked fully connected layer.

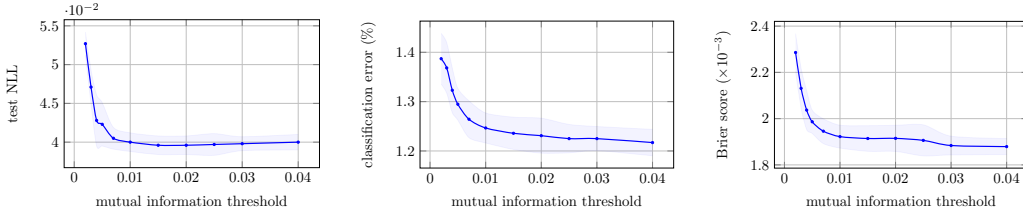

Figure 4: Results of an ablation study. The effect of conditioning the discriminative function on the generative process, $p(\phi|\theta)$ as measured by the test NLL, classification error, and Brier score. It is evident that performance worsens after weakening the dependence of $\phi$ on $\theta$.

## 4.2 Calibration

BRAINet can be perceived as a compact representation of an ensemble. We demonstrate on MNIST that it achieves higher classification accuracy and is better calibrated than Deep Ensembles for the same model size (Figure 5). Here, we used the simultaneous inference mode of BRAINet. Next, we evaluate the accuracy and calibration of BRAINet as a function of stochastic forward passes, and find it to significantly outperform Bayes-by-Backprop [1] and MC-dropout (Figure 6). We also find that using the BRAINet structure within either Deep Ensembles or with MC-dropout methods, further improves these later approaches. For that we use common UCI-repository [4] regression benchmarks. Results are reported: Appendix D-Table 1 for Deep Ensembles, and Appendix D-Table 2 for MC-dropout. Finally, we compare BRAINet to various state-of-the-art methods on large networks. In all benchmarks, BRAINet achieves the lowest expected calibration error [8] (Appendix D-Table 3).

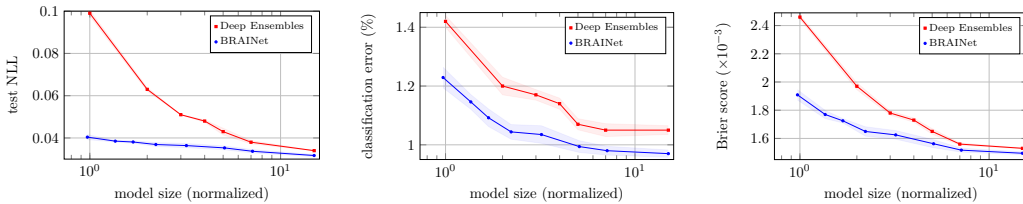

Figure 5: Performance as a function of normalized model size. Test NLL, classification error, and Brier score of BRAINet, compared to Deep Ensembles [16]. $X$-axis is the model size divided by the size of a single network (240K parameters) in the Deep Ensembles model. For BRAINet, up to model size 5, $s = 2$, and from model size 7 and above, $s = 3$. Different BRAINet sizes, for a given $s$, are obtained by varying the number of neurons in the dense-layers.

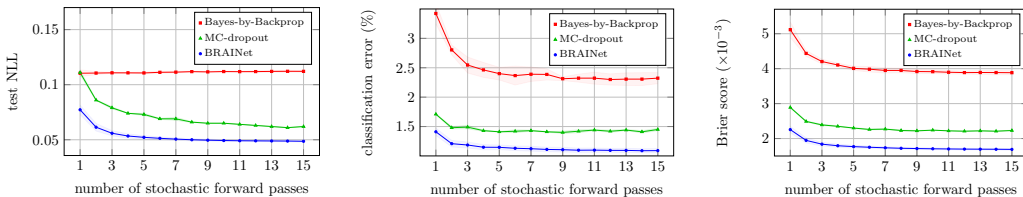

Figure 6: Performance as a function of the number of forward passes (number of sampled networks). Test NLL, classification error, and Brier score of BRAINet, compared to MC-dropout [5] and Bayes-by-Backprop [1]. We used a BRAINet with $s = 2$. Model size is 240K parameters for MC-dropout and BRAINet, and double for Bayes-by-Backprop.

## 4.3 Out-of-Distribution Detection

Next, we evaluate the the performance of detecting OOD samples by applying the BRAINet structure after feature-extracting layers of common NN typologies. First, we compare it to a baseline network and MC-dropout (We found MC-dropout to significantly outperform Bayes-by-Backprop and Deep

Ensembles on this task). In order to evaluate the gain in performance resulting only from the structure, we use the cross-entropy loss for parameter learning. Using other loss functions that are suited for improving OOD detection [3, 19], may further improve the results (see the next experiment). In this experiment we used a ResNet-20 network [9], pre-trained on CIFAR-10 data. For MC-dropout, we used a structure having 2 fully-connected layers, and for BRAINet, we learned a 2-layer structure with $s = 2$. Both structures have 16K parameters and they replace the last layer of ResNet-20. SVHN dataset [22] is used as the OOD samples. We calculated the area under the ROC and precision-recall curves, treating OOD samples as positive classes (Table 1).

Table 1: OOD detection (SVHN dataset) by replacing the last layer of ResNet20, pre-trained on CIFAR-10, with a learned structure. Parameters are trained using cross-entropy loss. Two inference mode of BRAINet: a single simultaneous forward pass (BRAINet sm.), and multiple stochastic forward passes. MC-dropout and BRAINet use 15 forward passes (see also Appendix D-Figure 3).

| | | AUC-ROC | | | | AUC-PR | | | |
|---|---|---|---|---|---|---|---|---|---|
| METHOD | ERR | MAX. P | ENT. | MI | E.ENT. | MAX. P | ENT. | MI | E.ENT. |
| BASELINE | 7.7 | 90.38 | 90.74 | — | — | 93.94 | 94.09 | — | — |
| BRAINET SM. | 8.0 | 91.62 | 92.18 | — | — | 94.91 | 94.80 | — | — |
| MC-DROPOUT | 8.2 | 90.73 | 90.26 | 84.74 | 90.61 | 94.31 | 93.71 | 88.89 | 94.46 |
| | ±0.1 | ±0.78 | ±0.81 | ±1.07 | ±0.93 | ±0.33 | ±0.34 | ±0.45 | ±0.46 |
| BRAINET | **7.5** | **92.13** | **92.61** | **91.87** | *92.98* | **95.37** | **95.36** | **94.6** | *95.64* |
| | ±0.1 | ±0.15 | ±0.03 | ±0.13 | ±0.10 | ±0.13 | ±0.07 | ±0.25 | ±0.05 |

Lastly, we demonstrate that training the parameters of BRAINet using a loss function, specifically designed for OOD detection [3], achieves a significant improvement over a common baseline [11] and improves state-of-the art results (Table 2).

Table 2: OOD detection by training BRAINet parameters using a loss function designed for OOD detection. A comparison between a baseline [11], confidence-based thresholding [3], and BRAINet with the same confidence measure. Architecture: VGG-13, in-distribution: CIFAR-10, OOD: TinyImageNet. BRAINet replaces the last layer. FPR @TPR=95%: false positive rate at true positive rate of 95%. Detection error: minimum classification error over all possible thresholds. AUC-ROC and AU-PR: area under ROC and precision-recall curves. "in"/"out" indicate that in/out-of-distribution data is the positive class. An arrow indicates if lower (↓) or higher (↑) is better.

| MEASURE | | BASELINE [11] | CONFIDENCE [3] | BRAINET |
|---|---|---|---|---|
| CLASSIFICATION ERROR | ↓ | **5.28** | 5.63 | 5.65 |
| FPR @TPR=95% | ↓ | 0.438 | 0.195 | **0.124** |
| DETECTION ERROR | ↓ | 0.120 | 0.092 | **0.076** |
| AUC-ROC | ↑ | 0.935 | 0.970 | **0.980** |
| AUC-PR (IN) | ↑ | 0.946 | 0.974 | **0.982** |
| AUC-PR (OUT) | ↑ | 0.917 | 0.965 | **0.979** |

## 5   Conclusions

We proposed a method for confounding the training process in deep neural networks, where the discriminative network is conditioned on the generative process of the input. This led to a new architecture—BRAINet: a hierarchy of deep neural connections. From this hierarchy, local sub-networks can be sampled proportionally to the posterior of local causal structures of the input. Using an ablation study, we found that even a weak relation between the generative and discriminative functions results in a significant gain in calibration and accuracy. In addition, We found that the number of neural connectivity patterns in BRAINet is adjusted automatically according to the uncertainty in the input training data. We demonstrated that this enables estimating different types of uncertainties, better than common and state-of-the-art methods, as well as higher accuracy on both small and large datasets. We conjecture that the resulting model can also be effective at detecting adversarial attacks, an plan to explore this in our future work.

## Footnotes

*All authors contributed equally.

[1] A CPDAG (complete partially directed acyclic graph), a Markov equivalence class encoding causal relations among $\boldsymbol{X}$, is learned from nonexperimental observed data.

[2] As the structures of $\tilde{\mathcal{G}}$ and $\mathcal{G}$ are identical, differing only in edge direction, we assume a one-to-one mapping from each latent node in $\tilde{\mathcal{G}}$ to its corresponding latent node in $\mathcal{G}$.

[3]Using a decomposable score, such as BDeu, the score of a CPDAG is $r = \sum_{i=1}^{n} r(X_i|Pa_i)$.

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
