[Supplementary Material · Appendix-Modeling Uncertainty by Learning a Hierarchy of Deep Neural Connections.pdf]

# Appendix: Modeling Uncertainty by Learning a Hierarchy of Deep Neural Connections

**Raanan Y. Rohekar, Yaniv Gurwicz, Shami Nisimov, Gal Novik**

Intel AI Lab

## A  BRAINet Structure Learning: A Detailed Description

In this section we provide a detailed description of Algorithm 1 and its three main stages. For more details, specific to Bayesian networks (not to be confused with Bayesian neural networks), refer to Pearl [4].

**Stage a.** First in line 7, a bootstrap sample $\mathbf{x}^*$ is created by sampling-with-replacement from the training data, $\mathbf{x}$ (non-parametric bootstrap). The bootstrap principle is a common approach used to approximate a population distribution by a sample distribution [1].

Next in line 8, the bootstrap sample $\mathbf{x}^*$ is used for learning a Bayesian network structure $\mathcal{B}^*$, with sparser connectivity than $\mathcal{B}$, such that $\mathcal{B}$ can mimic $\mathcal{B}^*$, $\mathcal{B}^* \preceq \mathcal{B}$ [4]. That is, for every set of parameters $\nu$, quantifying $\mathcal{B}$, there exists a set of parameters $\nu^*$, quantifying $\mathcal{B}^*$, such that $p_{\mathcal{B},\nu}(X) = p_{\mathcal{B}^*,\nu^*}(X)$. The $\mathcal{B}^*$ structure is learned by testing conditional independence of order $n$ between pairs of nodes connected in $\mathcal{B}$. That is, $X \perp\!\!\!\perp X'|\boldsymbol{S}$ for every connected pair $X \in \boldsymbol{X}$ and $X' \in \boldsymbol{X}_{\mathrm{ex}}$ given a condition set $\boldsymbol{S} \subset \{\boldsymbol{X}_{\mathrm{ex}} \cup \boldsymbol{X}\}$ of size $n$. Edges between conditionally independent nodes are then removed, and the remaining edges are directed by applying two rules. First, v-structures are identified and directed. Then, edges are continually directed, by avoiding the creation of new v-structures and directed cycles, until no more edges can be directed. Following the definition of Yehezkel & Lerner [6] for d-separation resolution, we say that this function increases the graph d-separation resolution from $n - 1$ to $n$.

Finally in line 9, the procedure `SplitAutonomous` (line 7) identifies autonomous sets: one descendant set, $\boldsymbol{X}_{\mathrm{D}}$, and $k$ ancestor sets, $\boldsymbol{X}_{\mathrm{A}1}, \ldots, \boldsymbol{X}_{\mathrm{A}k}$. This decomposition is achieved in two steps. First, the nodes having the lowest topological order (nodes without outgoing directed edges) are grouped into $\boldsymbol{X}_{\mathrm{D}}$, and then, $\boldsymbol{X}_{\mathrm{D}}$ is removed (temporarily) from $\mathcal{B}$ revealing unconnected sub-structures. The number of unconnected sub-structures is denoted by $k$ and the nodes set of each sub-structure is denoted by $\boldsymbol{X}_{\mathrm{A}i}$ ($i \in \{1 \ldots k\}$).

**Stage b.** The BRAINet structure learning algorithm is recursively called for each autonomous (line 11) and descendant (line 12) set. An autonomous set in $\mathcal{B}$ includes all its nodes' parents (complying with the Markov property) and thus, the algorithm can be called recursively and independently. Each recursive call returns a BRAINet model: $L_{\mathrm{A}i}$ for each ancestor set $\boldsymbol{X}_{\mathrm{A}i}$, and $L_{\mathrm{D}}$ for the descendant set $\boldsymbol{X}_{\mathrm{D}}$. Note that, each recursive call, has a smaller field-of-view (FOV) over the input $\boldsymbol{X}$ compared to its caller function ($\boldsymbol{X}_{\mathrm{A}i} \subset \boldsymbol{X}$).

**Stage c.** BRAINet models returned from the recursive calls are merged by connecting the deepest layer of each of them in the following manner. First in line 13, a layer container is created, denoted by $\boldsymbol{L}^t$, where $t$ is an index created in line 6 and represents one of $s$ bootstrap-created splits. Next in line 14, for the $i$-th BRAINet model, created for ancestor set $\boldsymbol{X}_{\mathrm{A}i}$, a layer of neurons, $L_i^t$ is created. Finally in line 15, the BRAINet models returned from the recursive calls are connected. The deepest layer of $L_{\mathrm{D}}$ is connected to all the newly created layers $L_1^t, \ldots, L_k^t$ and the deepest layer of the $i$-th BRAINet model created for $\boldsymbol{X}_{\mathrm{A}i}$ is connected to layer $L_i^t$.

As an example, in the diagram below, we show the results of lines 7-9 of Algorithm 1 in a tree-form (we use $s = 2$). Each rectangle encapsulates the result of multiple possible decomposition, obtained from multiple bootstrap samples. For simplifying the explanation, we omit the distinction between ancestor and descendant sets. For example, in the top rectangle, $X$ is decomposed, using statistical independence tests of order $n$, in two ($s = 2$) manners: 1) $\{X_1, X_2, X_3, X_4\}$, 2) $\{X_5, X_6, X_7, \}$. Each set $X_i$ is further recursively decomposed using statistical independence tests of order $n + 1$. For example, $X_7$ is decomposed in two manners (using different bootstrap samples).

## B    Complexity of BRAINet Structure Learning

The complexity of the proposed algorithm is essentially identical to that of the B-RAI algorithm. That is, $\mathcal{O}(n^k s^{k+1})$ conditional independence tests, and $\mathcal{O}(ns^k)$ Bayesian scoring function calls., where $s$ is the number of splits, $n$ is the number of input variables ($|X|$), and $k$ is the maximal order of conditional independence in the data.

The running-trace of RAI in the worst-case scenario is a single path in the GGT, and has a CI-test complexity of $\mathcal{O}(n^k)$, and the ratio between B-RAI and RAI is $\sum_{i=0}^{k} s^i$.

For the Bayesian scoring function, the complexity is $\mathcal{O}(ns^k)$, as only the leaves (when the exit condition is satisfied) are scored. Note that the worst-case scenario is the case where the true underlying graph is a complete graph, which is not typical in real-world cases. In practice, significantly fewer CI tests and scoring operations are performed.

## C    BRAINet Uncertainty Estimation

### C.1    A Relation between Generative Uncertainty and Predictive Uncertainty

During the construction of a BRAINet model, multiple connectivity patterns at each point in the network are learned using different bootstrap samples. Thus, when the epistemic uncertainty of $\theta$ is high, connectivity patterns learned from different bootstrap samples are likely to be dissimilar. However, when the epistemic uncertainty of $\theta$ lowers (e.g., for a larger training set), the connectivity patterns for any two bootstrap samples are more likely to be similar, at which point the aleatoric uncertainty dominates, and further reducing epistemic uncertainty (e.g., by adding more training data) will not result in a reduction of the number of unique connectivity patterns.

### C.2    An example for Epistemic Uncertainty Estimation using MI Criterion

In this section we repeat the experiment described by Smith & Gal [5]. Result is in Figure 2, where we also plot 12 images generated by a VAE: 6 having low MI score, and 6 having high MI score.

Figure 1: Number of unique structures (neural connectivity patterns) embedded in a single BRAINet ($s = 3$) for MNIST, as a function of the training set size. As the epistemic uncertainty increases (small training sets), more unique structures are automatically encoded in BRAINet, resulting in a broader prior over the network parameters. At the other end, as the training set size increases, the number of unique structures decreases and converges to a number greater than one, indicating the existence of an aleatoric uncertainty. Results are averaged over 5 experiments; error bars indicate standard deviation.

Figure 2: Epistemic uncertainty estimate using BRAINet ($s = 2$) as measured by mutual information, visualized on the latent space of a VAE on MNIST. In this experiment, as suggested by Smith & Gal [5], a VAE with latent space size 2 was trained and the axes are the values of these 2 latent variables. Brighter pixels correspond to a higher mutual information, i.e. a higher uncertainty. Images were then generated by setting values for each latent variable, 100 values in $[-10 : +10]$. Colored pixels correspond to the training data. As demonstrated, ambiguous images, $\{A, ..., F\}$, yield high epistemic uncertainties, as opposed to the clearer images of $\{G, ..., L\}$.

## D  Experiments

This section contains tables and figures referenced in the paper. See captions for a detailed explanation.

Firstly, we wish to emphasize an important property of the BRAINet structure, which allows us to sample networks that are significantly smaller than the overall structure. This leads to a significantly smaller computational cost, during training and inference, than other methods. For example, in a BRAINet for MNIST, the average number of operations (multiply and add) during inference is $\sim \mathbf{5.5\times}$ smaller than other methods (and has $\sim \mathbf{5.5\times}$ fewer parameters to train each epoch). Moreover, in contrast to MC-dropout, which samples neurons from each layer, BRAINet samples full layers, making it computationally efficient using common hardware. In all our experiments, we report only the size of the *full* BRAINet structure and ignore the sizes of the sampled networks.

Figure 3: Area under ROC and precision-recall curves as a function of the number of stochastic forward passes. It is evident that BRAINet achieves high AUC even for a samll number of forward passes, compared to MC-dropout. This result corresponds to the OOD-detection experiment described in Table 1 of the main paper. Architecture: ResNet20, in-distribution: CIFAR-10, OOD: SVHN.

Table 1: The effect of conditioning the discriminative function on a MAP estimation of a generative network (BRAINet) for Deep Ensembles [3]. Regression Benchmark datasets comparing RMSE and NLL.

| DATASET | RMSE | | NLL | |
|---|---|---|---|---|
| | ENSEMBLE | BRAINET+ENSEMBLE | ENSEMBLE | BRAINET+ENSEMBLE |
| CONCRETE | 5.46± 0.52 | **4.17± 0.59** | 2.92± 0.16 | **2.56± 0.13** |
| BOSTON HOUSING | 2.90± 0.71 | 2.17± 0.56 | 2.37± 0.25 | **1.88± 0.21** |
| POWER PLANT | 4.11± 0.16 | 4.10± 0.14 | 2.82± 0.03 | 2.8± 0.03 |
| YACHT | 3.09± 1.03 | **1.10± 0.36** | 2.25± 0.39 | **1.09± 0.14** |
| KIN8NM | 0.09± 0.00 | **0.08± 0.00** | -1.18± 0.02 | **-1.23± 0.03** |
| ENERGY | 0.95± 0.26 | 0.71± 0.10 | 1.16± 0.25 | 1.02± 0.13 |
| NAVAL PROPULSION PLANT | 0.00± 0.00 | 0.00± 0.00 | -3.55± 0.06 | **-3.68± 0.05** |
| WINE | 0.64± 0.04 | 0.59± 0.05 | 0.92± 0.08 | **0.76± 0.05** |
| PROTEIN | 4.62± 0.17 | **4.16± 0.15** | 2.79± 0.03 | **2.60± 0.09** |

Table 2: The effect of conditioning the discriminative function on a MAP estimation of a generative network (BRAINet) for MC-dropout [2]. Regression Benchmark datasets comparing RMSE and NLL.

| DATASET | RMSE | | NLL | |
|---|---|---|---|---|
| | MC-DROPOUT | BRAINET+DROPOUT | MC-DROPOUT | BRAINET+DROPOUT |
| CONCRETE | 4.83± 0.52 | **3.09± 0.78** | 2.92± 0.09 | **2.64± 0.12** |
| BOSTON HOUSING | 2.80± 0.52 | **1.90± 0.38** | 2.39± 0.15 | **2.16± 0.08** |
| POWER PLANT | 4.02± 0.18 | 4.00± 0.16 | 2.80± 0.05 | 2.79± 0.04 |
| YACHT | 1.42± 0.48 | **0.61± 0.25** | 1.60± 0.13 | **1.38± 0.06** |
| KIN8NM | 0.10± 0.00 | **0.08± 0.00** | -0.95± 0.03 | **-1.10± 0.03** |
| ENERGY | 0.88± 0.13 | **0.56± 0.08** | 1.62± 0.03 | **1.44± 0.02** |
| NAVAL PROPULSION PLANT | $(1.8± 0.2)e^{-3}$ | $(1.2± 0.3)e^{-3}$ | -4.22± 0.01 | **-4.35± 0.02** |
| WINE | 0.62± 0.04 | **0.54± 0.04** | 0.93± 0.06 | **0.82± 0.05** |
| PROTEIN | 3.60± 0.03 | **3.46± 0.19** | 2.69± 0.01 | 2.67± 0.03 |

Table 3: Comparison between BRAINet and various state-of-the-art methods on large networks. In all benchmarks, BRAINet achieves the lowest expected calibration error [9].

| DATASET | MODEL | SGD | SWA | SWAG-DIAG | SWAG | KFAC-LAPLACE | SWA-DROPOUT | SWA-TEMP | BRAINET |
|---------|-------|-----|-----|-----------|------|--------------|-------------|----------|---------|
| CIFAR-10 | VGG-16 | 0.0483 | 0.0408 | 0.0267 | 0.0158 | 0.0094 | 0.0284 | 0.0366 | **0.0090** |
| CIFAR-10 | PRERESNET-164 | 0.0255 | 0.0203 | 0.0082 | 0.0053 | 0.0092 | 0.0162 | 0.0172 | **0.0036** |
| CIFAR-10 | WRN28x10 | 0.0166 | 0.0087 | 0.0047 | 0.0088 | 0.0060 | 0.0094 | 0.0080 | **0.0040** |
| CIFAR-100 | VGG-16 | 0.1870 | 0.1514 | 0.0819 | 0.0395 | 0.0778 | 0.1108 | 0.0291 | **0.0247** |