[Reviews · NeurIPS 2019]

Reviewer 1



Here are my comments for the paper: - B2N, RAI, and GGT abbreviations are never defined in the paper; the have been just cited from previous works (minor). A short background section on these methods can also include their full name. - The distinctions and contributions of this work compared to [25] and [28] can be written in a more clear way. As far as I understand, the proposed method is B2N with B-RAI instead of RAI which was originally proposed in [25]. This allows the model to sample multiple generative and discriminative structures, and as a result create an ensemble of networks with possibly different structures and parameters. Maybe a better way for structuring the paper is to have a background section on B-RAI and B2N, and a separate section on BRAINet in which the distinction with other works and contribution is clearly written. - For OOD experiments, the prior networks loss function [18] is used but the method is not used as a baseline in the experiments. It would be nice to have the results of OOD experiments with [18] and possible some other more recent work on OOD detection. See for instance, “Reliable uncertainty estimates in deep neural networks using noise contrastive priors” or "A Simple Unified Framework for Detecting Out-of-Distribution Samples and Adversarial Attacks”. Specifically, the latter has results on the same dataset. - In terms of empirical evaluation and compared to much simpler baselines which do not require structure learning (e.g. [16]), the method doesn’t show a major improvement. Overall, I found the idea of modeling uncertainty with an ensemble of networks with possibly different structures and parameters interesting. However, given the complexity of the model which requires structure learning using a recursive algorithm, the performance gain is not that impressive. This weakness could have been outweighed by technical novelty, but the method is very similar to the B2N approach (with B-RAI instead of RAI) which in my opinion makes the novelty incremental. --------------------------------- I've read authors' feedback and I'd like to change my score; however, I'm still not convinced that the contribution is significant enough to pass the conference acceptance standard.

Reviewer 2



Summary The authors propose a new method for Bayesian deep learning. The idea is to learn a hierarchy of generative models for the data (using [1]), that can then be converted into discriminative models (using [2]). The authors then assign a posterior value to each of the subnetworks, and sample subnetworks to perform Bayesian model averaging. The authors then conduct a series of experiments on different types of uncertainty estimation with strong results. I have limited familiarity with probabilistic graphical models, and I am unable to provide feedback on the quality of the methodological contribution in the paper. For this reason, in my review I focus on the empirical evaluation and clarity. Originality The paper provides a very non-standard approach to Bayesian inference for deep neural networks. The method is very original and interesting. Related work on Bayesian deep learning is cited appropriately. Clarity I believe some parts of the paper could be clarified: 1. It was unclear to me how exactly is the method applied in the experiments. From Section 3 my understanding was that the method learns a hierarchy of Bayesian neural networks from scratch. How is it combined with given neural network architectures, e.g. in Table 1? Is it applied to learn the last several layers of the neural network? 2. Is it correct that the proposed method proceeds as follows: First, it produces a hierarchy of architectures for some subset of a given neural network architecture (see question 1). Then, it trains the weights in this architecture by sampling sub-networks uniformly and updating the weights by SGD. Then the method performs Bayesian model averaging with one of the two strategies described in the paper. I would recommend to include a high-level outline like this in the camera-ready version. 3. In section 3.2.2, what is c? Is it the score produced by B-RAI for the sub-network? If so, wouldn’t it make more sense to re-weight sub-networks based on the corresponding training loss value in the posterior? Quality As I mentioned above, I will mostly focus on the empirical evaluation and results. The reported results are generally very promising. 1. Section 4.2: the authors compare the predictive accuracy and likelihoods for the proposed method against deep ensembles, Bayes-by-backprop and Deep Ensembles on MNIST. How exactly were the Deep Ensembles trained here, keeping the model size fixed? 2. While the accuracy and NLL results appear strong, it would also be interesting to see calibration diagrams and ECE reported, as in [3] and [4]. 3. In Section 4.3 the authors apply the proposed method to out-of-distribution (OOD) data detection. The authors achieve performance stronger than other baselines on identifying SVHN examples with a network trained on CIFAR-10. For the dropout baseline, have you tried to apply it before every layer rather than just the last 2 layers? 4. The results in Table D-3 show expected calibration error for BRAINet on a range of modern architectures on image classification problems. Could you please report accuracies or negative likelihoods (or both) at least for a subset of those experiments? While ECE is a good metric for the quality of uncertainty, it is possible to achieve good ECE with poor predictive accuracy, so a combination of the two metrics would give a more complete picture of the results. Minor issues: Line 56: add whitespace before “In”. Lines 93-94: this sentence is not clear to me. Line 212: “outperforms” -> “outperform”. Significance The paper proposes a new ingenious approach to Bayesian deep learning. The reported results seem promising. Importantly, the method is applicable to modern large-scale image classification networks. However, the clarity of the paper could be improved. [1] Bayesian Structure Learning by Recursive Bootstrap; Raanan Y. Rohekar, Yaniv Gurwicz, Shami Nisimov, Guy Koren, Gal Novik [2] Constructing Deep Neural Networks by Bayesian Network Structure Learning; Raanan Y. Rohekar, Shami Nisimov, Yaniv Gurwicz, Guy Koren, Gal Novik [3] A Simple Baseline for Bayesian Uncertainty in Deep Learning; Wesley Maddox, Timur Garipov, Pavel Izmailov, Dmitry Vetrov, Andrew Gordon Wilson [4] On Calibration of Modern Neural Networks; Chuan Guo, Geoff Pleiss, Yu Sun, Kilian Q. Weinberger

Reviewer 3



[Update after author feedback: I have read the other reviews and the author feedback and want to thank the authors, particularly for answering my clarifying questions in detail. As the other reviewers pointed out as well, the paper can still be improved in terms of clarity, particularly when addressing the (deep) neural network community as a main target audience. I want to encourage the authors to try and improve clarity as much as possible. To me personally, the paper is promising and interesting, but I find it hard to judge how novel the proposed algorithm is (hence my confidence of 2). I am personally leaning towards acceptance, though I understand the criticism and (as the score of 7 implies) "would not be upset if the paper is rejected".] The paper proposes to learn the structure (connectivity structure and parameter sharing) of a stochastic ensemble of neural networks. This allows for solutions ranging anywhere from large connectivity variation with full parameter sharing (MC Dropout) to solutions with no connectivity variation and no parameter sharing (“classical” ensemble of networks). To achieve this, the structure of an ensemble of networks is expressed as a prior over parameters in a Bayesian neural network (a common interpretation of both MC Dropout and deep ensembles). Crucially, the paper proposes to learn the structure, i.e. the prior, from the input data (e.g. images in image classification) in a self-supervised fashion. The claim is that such a prior (now actually a conditional distribution over parameters given the input data), if tuned to the generative process of the input data, reduces unwarranted generalization for out-of-distribution data of the “prior” (the conditional distribution) over the structure. To learn a prior that is tuned to the generative process of the data, the paper applies a recently proposed Bayesian structure learning algorithm (B-RAI) that induces a distribution over discriminative network structures that are proved to mimic the generative structure (line 115 to 117), thus yielding a dependency between the structure of an implicit generative model of the data and the structure of the discriminative model (line 127 to 128). The paper explains and illustrates the algorithm (both training and inference) and shows how to use it for estimating predictive uncertainties. Finally the paper evaluates the performance of the proposed algorithm by investigating calibration of uncertainty estimates (on MNIST, as well as CIFAR-10/-100 in the appendix) and out-of-distribution detection on (SVHN and CIFAR-10). The method performs well compared to MC Dropout and Deep Ensembles, and some other state-of-the-art methods for calibration (in the appendix). Combining the method with MC Dropout or Deep Ensembles yields even better performance. I should point out that I found the paper somewhat hard to follow and I might have missed something important or misunderstood small or large aspects of the paper. Therefore I want to encourage the authors to comment/correct my summary of the paper if necessary. I should also point out that I had reviewed this paper at a previous venue and was happy to see that many of the previous reviewer’s comments were taken into account, particularly w.r.t. to improving clarity and improving the empirical comparisons. Before stating my personal opinion, I would like to ask the authors a series of clarifying questions: 1) The paper claims that the method prevents unwarranted generalization on out-of-distribution data - under what conditions does that claim hold? Does the claim essentially rest on p(\phi|\theta) being “well behaved” for out-of-distribution data? 1a) If yes to the latter, does that mean that for out-of-distribution data p(\phi|\theta) needs to spread probability mass across \phi, i.e. different structures - which corresponds to having high uncertainty over the structure for out-of-distribution data? 1b) If yes to 1a), what are the arguments for why this is likely to hold in practice and can you show some empirical results that support these arguments? 2) Would it be fair to say that preventing unwarranted generalization for out-of-distribution data could in principle also be achieved with a large ensemble of networks of different (connectivity) structure, i.e. no parameter sharing (to prevent unwarranted generalization due to learned parameters) and a large variety of structure (to prevent unwarranted generalization due to a fixed connectivity structure)? 2a) If yes to 2), would the advantage of the proposed approach be that the method captures an (infinite) ensemble of networks of different structure more efficiently (in computational terms) in the form of a Bayesian neural network? Assuming that my current understanding of the paper is mostly correct (i.e. my summary is mostly correct and most of the questions above are answered positively), I am slightly in favor of accepting the paper, since it addresses two timely issues (well calibrated predictive uncertainties, and a formulation for Bayesian neural networks where the connectivity structure is not fixed but drawn from a learned prior) in a novel and original way. I think that the current experimental section is solid and convincing enough for publication, though it could always be improved of course. To me the paper has reached a point where it would be interesting to see independent replications and discussion as well as extensions of the findings by the wider community. I want to encourage the authors to try and further improve the clarity of the manuscript and presentation of the method - with a particular focus on (Bayesian) deep learning practitioners (see some suggestions for improvement below). Since there is still quite some uncertainty in my current understanding of the paper, I also want to point out that it is quite possible that I’ll change my final verdict quite drastically based on the author feedback and other reveiwer’s comments. Originality: BRAINet structure learning algorithm: medium to low (I find it hard to judge whether the algorithm follows trivially from previous work, or requires major innovation), improving calibration and out-of-distribution detection with the method: medium to high Quality: Medium - there is no thorough discussion of related work, the paper does not mention (current) limitations and shortcomings of the approach, but there is a large number of empirical evaluations with interesting comparisons against state-of-the-art approaches Clarity: Low - though clarity has certainly improved since the last version of the manuscript, I still had a hard time to distill the main idea, and the experimental details w.r.t. how exactly BRAINet is applied and trained still remain a bit muddy (without looking at the code). Significance: Medium - the approach is novel and original and the experiments look promising. To be fully convinced, I’d like to see a thorough discussion of the shortcomings, how well the approach scales potentially and results on larger-scale tasks (I do acknowledge that not all of this is necessarily within the scope of a single paper) Minor comments: Tuning the prior to the generative process of the data and then using it as a prior for the discriminative model sounds like a mild version of an Empirical Bayesian method, which are known to break some appealing properties of proper Bayesian methods at least in theory. While I am personally happy with exploring Empirical Bayesian methods (in fact many modern Bayesian neural network algorithms, for instance some Bayesian neural network compression methods based on sparse Bayesian learning, learn the prior and thus also fall into the category of Empirical Bayesian methods), I would not necessarily over-emphasize the Bayesian nature of the proposed approach. It could potentially be very interesting compare and connect/apply BRAINet to Graph Neural Networks. While a full treatment of this is beyond the scope of this paper, at least some discussion about similarities and differences in the approaches and whether they can easily be combined would be interesting.

[Author Response · NeurIPS 2019]

**To all reviewers.** We would like to sincerely thank you for your important ideas and constructive comments. First, we would like to
clarify that B-RAI [24] is a recently proposed algorithm for estimating the posterior *probability* of causal relations among observed
variables. It is not related to the deep learning domain. The B2N algorithm [25], introduces principles for converting *fixed* causal
relations into a deep neural network (NN) structure. Simply using B-RAI in B2N is not trivial, and we introduced three important
algorithmic contributions: a principled method for weight sharing among ensemble-networks (Algorithm 1), a scheduling algorithm
for parameter learning in Sec. 3.2.2 (a method for jointly sampling parameters for update), and an anytime stochastic inference &
uncertainty estimation (e.g., expected entropy) in Sec. 3.2.3. In addition, a theoretical contribution is the suggestion of a lurking
confounder (Fig. 1), and an approach for integrating it out (Eq. 6). Thus, making the prior distribution over parameters of the
discriminative NN dependant on the underlying generative mechanisms of unlabeled data. Importantly, this confounder is *missing*
(and untreated) from traditional Bayesian deep learning approaches, and to the best of our knowledge, our work is the first to address
it. We will clearly state these contributions in the paper. In our experiments we distilled the contribution of the BRAINet structure
and training & inference algorithms. Distilling was done by an ablation study, and the use of a standard loss function for OOD and
calibration experiments (we also demonstrated training BRAINet with OOD-detection loss function). We added all required answers
and clarifications, which will improve the quality of the final version, and appreciate if you will up your scores accordingly.

**To reviewer #1.** As you suggest, we will define B2N, RAI, and GGT in the paper. BRAINet demonstrates a clear advantage over
Deep-Ensembles in Fig. 5, and over MC-dropout & Bayes-by-Backprop in Fig. 6. A clear advantage in both classification accuracy
and calibration. Shaded areas represent $2\sigma$. An ensemble of 15 (last point on the curve, Fig. 5), having a total of 3.6M parameters, is
equivalent or worse than BRAINet with 3X fewer parameters (6th point on the graph). We will use a different scale for the axes in Fig.
5 to improve clarity. Compared to MC-dropout and Bayes-by-Backprop, for the same model size, BRAINet with 2 forward passes
significantly outperforms 15 forward passes in MC-dropout and Bayes-by-backprop. Furthermore, a forward pass in MC-dropout
uses most of the network weights ($\geq 50\%$), whereas BRAINet uses a fraction of it ($<10\%$). OOD Detection: other baselines, such as
[3][18], rely on defining a loss function, specifically crafted for OOD detection, often using some OOD data during training [18].
Optimizing for a specific loss hinders other objectives, e.g., accuracy and calibration. In all our experiments we used the common
cross-entropy loss, except for the results in Table 2 where we used [3] to demonstrate that BRAINet can be coupled with other
methods to further improve their performance. BRAINet is not targeted specifically at OOD detection, and in fact may even improve
accuracy and calibration in addition to being able to detect OOD (Table 1). In this manner, BRAINet is similar to other ensemble and
Bayesian NN methods against which we compared. Thank you for your comments, we will improve the clarity accordingly.

**To reviewer #2.** Clarity section: **(1)** Our method adds predictive uncertainty to existing topologies by using their feature extractors
(embeddings) and replacing their heads. Rohekar et al., (2018) demonstrated that the B2N algorithm can replace the last several
layers (convolutional and MLP) of existing networks while maintaining accuracy. Relying on their findings, we followed the same
practice but added a new capability: (anytime) uncertainty estimation. We will clarify this in the experiments section. **(2)** You are
correct. As you proposed, we will include the proposed high-level outline. **(3)** In section 3.2.2, $sc$ is the score produced by B-RAI. It
is the log of the posterior probability of the structure. This allows marginalizing out $\theta$ (approximating Eq. 6). Quality section: **(1)**
Yes. The model size of a single network is fixed $240K$, with 200 neurons in each layer. The difference along the X-axis (Fig. 5) is
due to a varying ensemble size. A sampled network from BRAINet is $\geq 240K$, and the max number of neurons in a layer is $\leq 200$.
We will include these details in the appendix. **(2)** ECE and calibration diagrams for section 4.2 are under work (currently a clear
advantage of BRAINet) and will be included in the appendix. **(3)** No. We have not tried applying dropout before every layer. This
can further improve the results for both BRAINet (adding dropout to the feature extraction layers) and MC-dropout. We preferred a
setup in which we can distill the effect of BRAINet without other factors. We will clarify in the paper that further improvements can
be made using you suggestion. **(4)** As requested, here is the table. Thank you for identifying typos.

| DATASET | MODEL | SGD NLL/ACC | SWA NLL/ACC | SWAG-DIAG NLL/ACC | SWAG NLL/ACC | KFAC-LAPLACE NLL/ACC | SWA-DROPOUT NLL/ACC | SWA-TEMP NLL/ACC | BRAINET NLL/ACC |
|---|---|---|---|---|---|---|---|---|---|
| CIFAR-10 | VGG-16 | 0.3285/93.17 | 0.2621/93.61 | 0.22/93.66 | 0.2016/93.60 | 0.2252/92.65 | 0.2328/93.23 | 0.2481/93.61 | **0.2011/93.81** |
| CIFAR-10 | PRERESNET-164 | 0.1814/95.49 | 0.145/96.09 | 0.1251/96.03 | **0.1232**/96.03 | 0.1471/95.49 | 0.127/**96.18** | 0.1347/96.09 | 0.1245/95.90 |
| CIFAR-10 | WRN28x10 | 0.1294/96.41 | 0.1075/96.46 | 0.1077/96.41 | 0.1122/96.32 | 0.121/96.17 | 0.1094/96.39 | 0.1064/96.46 | **0.1044/96.48** |
| CIFAR-100 | VGG-16 | 1.7308/73.15 | 1.278/74.30 | 1.0163/74.68 | 0.948/74.77 | 1.1915/72.38 | 1.1872/72.63 | 1.0386/74.30 | **0.0935/74.96** |

**To reviewer #3.** Answers to the clarifying questions: **(1)** Yes. For in-distribution, $\phi$ is conditionally independent of $X$ given $\theta$, i.e.,
$p(\phi|\theta, X_{\text{in}}) = p(\phi|\theta)$. For OOD this does not hold, $p(\phi|\theta, X) = p(\phi|\theta, X_{\text{OOD}})$. **(1a)** Yes, for OOD $p(\phi|\theta, X_{\text{OOD}})$ is expected to
spread the probability mass across $\phi$ due to the direct dependency of $\phi$ on arbitrary $X_{\text{OOD}}$ (which is also the case for in-distribution
in common methods, e.g., MC-dropout, missing the $\theta$ confounder). **(1b)** In practice, this results in inconsistent responses by the
sampled networks, i.e., $p(Y|X_{\text{OOD}})$ is expected to be spread. **(2)** Yes. That is accurate with one addition. Our method estimates the
(posterior) probability of each structure (a score) given unlabeled data ($X$). This leads to Bayesian model averaging, where each
structure is weighted differently. **(2a)** In the space of structures, the posterior distribution (given $X$, a prior for the parameters) may
have multiple modes. We wanted to sample structures from this distribution; however we further wanted to exploit any structural
similarities (e.g., distinct structures sampled from the same mode), and couple them into a hierarchy. This led to parameter sharing.
In our method, considering the deeper layer is equivalent to looking at this space with less resolution, blurring the modes thereby
merging close modes. Therefore, deeper layers share parameters. Layers closer to the input have distinct structures accounting for
the higher resolution view of this space. Thus, as you commented, BRAINet captures this space efficiently, in addition to it being the
only method that samples structures from this space ($p(\theta|X)$). Answers to the Improvements section: 1-**(I)** will be clarified based
on you clarifying questions in the previous section. And your suggestions for specific lines in the text. 1-**(II)** We will clarify that
the head of existing networks is learned while using their feature extraction layers. **(2)** As suggested, we will improve the related
work section. NN structure learning from the previous wave (late 80's early 90's) were mostly heuristic, greedy search algorithms.
Importantly, researcher of that wave didn't enjoy the advances in causal discovery (90's) and probabilistic machine learning on which
our work relies. **(3)** Shortcomings of our method: currently it is suitable for learning structures for features rather than pixels. **(4)** In
general, in all experiments we used default hyper parameters, with a fixed learning rate. We will provide all the required details.

[Meta-Review · NeurIPS 2019]

This paper proposes BRAINet as to combine Bayesian structure learning and Bayesian neural networks. In detail, the method assumes a confounder on the input features X and the discriminative network parameters \phi, where this confounder is defined as the generative graph structure on X, and the discriminative network shares the same structure as the generative one. Given observations X and Y, the approach first sample the generative graph structure from the posterior given X, then train the parameters of the corresponding discriminative network in order to fit the posterior distribution of phi given X and Y. Experiments are performed on calibration and OOD tasks, with MC-dropout and deep Ensembles as the main comparing baselines. Reviewers include experts in Bayesian structure learning and Bayesian neural networks. They read the author feedback carefully and engaged in post-rebuttal discussion actively. The author feedback addressed some of the confusions from the reviewers, and some of them increased the score to vote for acceptance with moderate/low confidence. However the main issues still need to be addressed: 1. Clarity: the current form of the paper is hard to follow for people from (Bayesian) deep learning community, which is the community that the paper seems to target for; 2. Novelty: the algorithm is an extension to the B2N/B-RAI approaches to deep neural networks; 3. Comparisons with other BNN methods are missing. Prior design and OOD task are two important and trending topics in Bayesian deep learning, so this paper provides a timely and potentially important contribution to the field. However after a brief read through of the paper, I agree with the reviewers that the clarity of the presentation needs to be improved. In particular, I think the big picture of the whole pipeline is not discussed in a clear way, and an algorithm describing the whole process of sampling graph then training \phi would be much helpful. Also the paper will benefit from a clear description of the novel contribution as compared to B2N/B-RAI. Finally I hope the authors can add in the extra experiments (some of them provided in author feedback) to the camera ready.